# Assessing Transcriptional Responses to Light by the Dinoflagellate *Symbiodinium*

**DOI:** 10.3390/microorganisms7080261

**Published:** 2019-08-14

**Authors:** Bahareh Zaheri, Steve Dagenais-Bellefeuille, Bo Song, David Morse

**Affiliations:** 1Institut de Recherche en Biologie Végétale, Département de Sciences Biologiques, Université de Montréal, Montréal, QC H1X 2B2, Canada; 2Agricultural Genomics Institute at Shenzhen, Chinese Academy of Agricultural Sciences, Shenzhen 518124, China

**Keywords:** dinoflagellate, transcriptional control, light regulation

## Abstract

The control of transcription is poorly understood in dinoflagellates, a group of protists whose permanently condensed chromosomes are formed without histones. Furthermore, while transcriptomes contain a number of proteins annotated as transcription factors, the majority of these are cold shock domain proteins which are also known to bind RNA, meaning the number of true transcription factors is unknown. Here we have assessed the transcriptional response to light in the photosynthetic species *Symbiodinium kawagutii*. We find that three genes previously reported to respond to light using qPCR do not show differential expression using northern blots or RNA-Seq. Interestingly, global transcript profiling by RNA-Seq at LD 0 (dawn) and LD 12 (dusk) found only seven light-regulated genes (FDR = 0.1). qPCR using three randomly selected genes out of the seven was only able to validate differential expression of two. We conclude that there is likely to be less light regulation of gene expression in dinoflagellates than previously thought and suggest that transcriptional responses to other stimuli should also be more thoroughly evaluated in this class of organisms.

## 1. Introduction

Dinoflagellates are protists with an unusual chromatin structure [1]. Dinoflagellate chromosomes are permanently condensed, and can be observed with light microscopy using fluorescent DNA stains such as DAPI or propidinium iodide [2]. When observed using the electron microscope, individual chromosomes display a characteristic whorled banding pattern reminiscent of the bacterial nucleoid [3], and nucleosomes have never been observed [4]. The unusual chromatin structure has a number of molecular correlates. The histone proteins are at very low levels [5], and while one or two histones have been detected in several species [6,7], all four core histones have not yet been detected in any species. Instead of histones, dinoflagellates are thought to compact their DNA with a high level of divalent cations [8], histone-like proteins (HLP) [9], and a dinoflagellate/viral nucleoprotein (DVNP) [6].

The unusual dinoflagellate nuclear structure raises problems with respect to the mechanisms of both DNA replication and transcription. Little is known about replication, but many studies have examined changes in gene expression in response to light. Some of these studies use qPCR to examine specific genes. For example, rhodopsin in *Prorocentrum* was followed over a 14:10 L:D cycle and was observed to vary three-fold between LD 0 and LD 14 [10]. Similarly, transcripts encoding the oxygen evolving enzyme OEE1 in *Symbiodinium* were 2.5 fold more abundant at LD 12 than at LD 0 [11], while transcript levels encoding the large rubisco subunit *rbcL* were three fold higher at LD 12 than LD 0 [12], suggesting higher levels of transcription during the light. Levels of the thylakoid chlorophyll a-chlorophyll c2-peridinin-protein-complex (acpPC) were reported to be higher in dark phase than in light phase [13], suggesting that lack of light promotes expression of the light harvesting gene transcript.

Other experimental approaches have used high throughput expression measures such as microarrays or RNA-Seq. One of the earliest studies on differential transcription between day and night was carried out with *Pyrocystis* using microarrays programmed with about 3500 cDNAs [14]. About 80 differentially expressed genes (DEG) (~2%) were found to have a >2-fold difference between day and night in this species, with a maximum observed change of 2.5-fold. A similar microarray study comparing genes expressed during the day and night in *Karenia brevis* found 458 DEG among the 4629 genes examined (10%), with a significance threshold of *p* < 0.0001 and >1.7-fold change [15]. RNA-Seq studies in *Symbiodinium microadriaticum* found 67 DEG (0.1%) between day and night using DESeq with a false discovery rate (FDR) of 0.1 [16] and a maximum fold change of 160. A much more substantial number of DEG were noted in a study using *Symbiodinium* strain SSB01 24 h after a transfer from light to dark [17]. There were 1334 DEG (2.2%) when cells were grown phototrophically and 1739 DEG (2.9%) when cells were grown mixotrophically. These studies used duplicates (phototrophic growth) or triplicates (mixotrophic growth), but instead of an FDR = 0.1, the cutoff values for significance were *p* < 0.05 and a >1.5-fold change. Lastly, 131 DEG (0.17%) were found when samples of *Lingulodinium polyedra* taken every six hours were compared using an FDR of 0.1 [18], but northern blots analyses of a random selection of these showed no changes suggesting all were likely to be false positives.

The initial goal of our experiments was to identify a light regulated gene in *S. kawagutii*, so that potential regulatory elements in the promoter could be determined from the genome sequence [19], dissected, and the potential transcription factors involved identified. In one approach, we selected three genes whose transcripts had been previously been reported to be light regulated in *Symbiodinium*, and verified their expression levels using northern blots. In a second approach, we analysed global transcript levels at dawn and dusk by RNA-Seq. However, neither of these approaches successfully identified a light regulated gene, consistent with what has been observed with the dinoflagellate *L. polyedra*. This suggests that previous reports of light responsive genes may have overestimated their number, and further suggests that other reports of transcriptional responses may also benefit from additional verification.

## 2. Materials and Methods

### 2.1. Cell Cultures

*Symbiodinium kawagutii* (CCMP2468) was obtained from the National Center for Marine Algae and Microbiota (Boothbay Harbor, ME, USA) and cultured at 24 °C under a 12:12 light:dark cycle (40 µE m^−2^ s^−1^) in standard f/2 medium lacking silicate [20]. *S. kawagutii* has recently been renamed *Fugacium kawagutii* [21].

### 2.2. Microscopy

Cells were concentrated by centrifugation, then resuspended in a solution of 3% freshly made formaldehyde in seawater for 10 min then washed three times with fresh seawater. Cells were finally resuspended in phosphate buffered saline containing 0.05% Tween 20 and 1 µg/mL propidium iodide for 30 min. Images were taken using a Zeiss confocal microscope using a 63× objective in green (PI) and red (chlorophyll) channels. 3D reconstructions were made using Fiji [22].

### 2.3. RNA Extraction and Northern Blots

For the high light condition, *S. kawagutii* cells in fresh normal culture medium were transferred to 350 μmol of photons m^−2^ s^−1^ high light (HL) for 24 h. *S. kawagutii* cells were harvested from LD0 (beginning of light), LD12 (beginning of darkness), and HL (24 h in constant light) cultures. Total RNA was extracted with Trizol as described [23], the quantity and quality assessed by spectrophotometry and then stored at −80 °C. *S. kawagutii* RBCL, AcpPC, OEE1, and Actin sequences were acquired from the genome sequence (http:web.malab.cn/symka_new). Primers were used to amplify the sequence from a first strand cDNA reaction product using *S. kawagutii* total RNA using the ProtoScript First Strand cDNA synthesis kit (Invitrogen, Burlington, ON, Canada). The identity of all PCR products was confirmed by sequencing.

Northern blotting analysis was performed as described [18], 10 µg total RNA was electrophoresed on a denaturing agarose gel. The RNAs were transferred onto a nylon membrane (HybondTM-H+; Amersham Pharmacia Biotechnology, Piscataway, NJ, USA) and cross-linked by UV. PCR generated probes were labeled with [α-32P] ATP (BLU512H, Perkin Elmer, Woodbridge, OA, Canada) for hybridization. Membranes were hybridized at 65 °C for 16 h and were then washed twice at 65 °C for 15 min. The radiolabeled membranes were exposed to a phosphoscreen for 24 h and revealed by Typhoon Imager.

### 2.4. RNA Sequencing

Quality control, library construction, and Illumina sequencing were performed on RNA samples prepared in triplicate from *S. kawagutii* at LD 0 and LD 12 at McGill University and Genome Quebec Innovation Centre (Montreal, QC, Canada). Between 36 and 57 million paired end reads were recovered for each of the six samples. Raw sequence reads are available from NCBI using the accession number PRJNA517819.

The unigene list used for read mapping was downloaded from the *S. kawagutii* genome resources (http://web.malab.cn/symka_new/). This unigene list, containing 70,987 sequences, as well as the six sets of paired-end Illumina sequence reads, were uploaded to the Galaxy web platform at usegalaxy.org. The reads were trimmed using TrimGalore and read counts for all sequences in the unigene list were determined using Salmon [24]. Statistical significance was estimated using DESeq2 running in R [25].

### 2.5. Quantitative PCR

cDNAs were prepared from *S. kawagutii* RNAs extracted from the cells collected at four-hour intervals over an LD cycle plus high light cultures using ProtoScript First Strand cDNA Synthesis Kit and an oligo(dT) primer (Invitrogen, Burlington, ON, Canada). Specific primers were designed for SymkaALLUN13501, SymkaALLUN19088, SymkaALLUN64909, and Actin. qPCR analysis was performed in a ViiA7 Real-Time PCR System (Applied Biosystem, Burlington, ON, Canada) using SYBR green qPCR Master Mix (Thermo Fisher, St. Laurent, QC, Canada). Gene specific primers (250 nM) and cDNA (150 ng) were used in a total volume of 10 µl. Triplicate samples from each of three biological replicates amplified using 10 min at 95 °C, followed by 35 cycles of 15 s at 95 °C, 1 min at 60 °C, and 35 s at 68 °C, followed by a melt curve stage from 60 °C to 95 °C to verify the absence of non-specific amplification.

For gene expression analysis, cycle threshold (Ct) values were obtained from the ViiA7 Real-Time PCR software (Thermo Fisher, St. Laurent, QC, Canada). Student’s *t*-test was used to verify the statistical significance of the data.

## 3. Results

*S. kawagutii* has a typical dinoflagellate chromosome structure. Cells at all times have visibly condensed chromosomes (Figure 1) that appear superficially similar to mitotic chromosomes in other cells. This compact structure suggests that transcription is likely to be challenging, since more typical eukaryotic cells transcription rates decrease during mitosis when the chromatin is more condensed [26].

In a first attempt to identify light responsive genes in *S. kawagutii*, examples were selected from the literature. We selected oxygen evolving enzyme (OEE1) where transcript levels changed in abundance by 2.5-fold between LD 0 and LD 12 [11], the large rubisco subunit *rbcL* where transcript levels were three-fold higher at LD 12 than LD 0 [12], and the thylakoid chlorophyll a-chlorophyll c2-peridinin-protein-complex (acpPC) where transcript levels were roughly three-fold higher in dark phase than in light phase [13]. Actin was chosen as a reference because it is not regulated by light in *Lingulodinium* [18] or as shown here by RNA-Seq in *S. kawagutii*. We amplified probes for these sequences from *S. kawagutii*, and used the probes to asses transcript levels at four-hour intervals over an LD cycle, as well as a culture left under high light conditions. In no case were different transcript levels observed (Figure 2). We conclude there is no support for the hypothesis that transcription of these three genes responds to light.

As a second attempt to identify light responsive genes, we prepared RNA samples in triplicate from *S. kawagutii* at LD 0 (dawn) and LD 12 (dusk). We reasoned that any light responsive genes would accumulate during the light period, and these would thus have higher levels at the end of the light phase. We compared read counts using the DESeq with a Benjamini–Hochberg correction (FDR = 0.1) to determine significant changes. A total of 7 changes (0.01%) were observed, all with higher levels at LD 0 than at LD 12 (Figure 3). Since all seven were higher at LD 0, this suggested that if these were truly light-regulated genes they would be induced by darkness or inhibited by light. These seven sequences were identified by BLAST searches (Table 1), and none correspond to the three sequences tested by northern blots. When the stringency of statistical significance was increased by setting the FDR to 0.05, only one of these was observed to display a statistically significant change. When a Bonferroni correction was applied instead of the Benjamini–Hochberg correction, four genes showed significant changes with *p* < 0.05, and one with *p* < 0.01. We conclude the number of significant changes in transcript levels is very low.

To validate the differential expression of the seven genes detected by RNA-Seq, we performed qPCR to assess the relative levels of three randomly selected genes (SymkaALLUN13501, SymkaALLUN19088, and SymkaALLUN64909). Assays were performed in triplicate for each of three biological replicates, and only two of these (ALLUN13501 and ALLUN19088) showed a significant difference between the two times (Figure 4). Since lower Ct values reflect higher transcript levels (i.e., transcript levels for these two genes are slightly higher at LD 0, as also found by RNA-Seq), we conclude that at least some of the seven genes with different levels as measured by RNA-Seq may reflect real differences in transcript levels. We note, however, that the fold difference appears smaller than that predicted by RNA-Seq.

Finally, to gain a global picture of the different fold changes detected, significant or not, we plotted the number of times different fold changes were observed as a function of the fold change (Figure 5). This analysis reveals a normal distribution of fold changes within the data set. To test the symmetry of the bell curve, positive fold changes were plotted as a function of negative fold changes (Figure 5 inset). The resulting curve is essentially a straight line with a slope of −1. The few exceptions to the linear relationship do not correspond to the genes classified as significant by DESeq. We conclude there is no overall bias for either positive or negative changes in transcript abundance between the two times examined.

## 4. Discussion

In many of the studies reporting differential gene expression as a result of light, only a single method was used to measure transcript abundance. For example, qPCR, northern blots, microarrays, or RNA-Seq have been used in individual studies but were not, with few exceptions, combined in the same study. One notable exception in *Lingulodinium* first used RNA-Seq to identify DEG and then verified a random selection of these using northern blots. Since northern blots failed to confirm the RNA-Seq-derived DEG, it was concluded all were likely to be false positives. This underscores the importance of validating high-throughput approaches, and suggests that it would be beneficial when several methods are combined to test for DEG.

The RNA-Seq experiments reported here used DESeq2 to identify DEG, with the threshold for significance determined by a false discovery rate (FDR) of 0.1. The FDR method, developed by Benjamini and Hochberg, uses a statistical method to restrain the number of false positives to a fixed percentage of the total positives, and thus provides increased confidence that significant changes are in large datasets are likely true positives [27]. The FDR can be thought of as a method for using lower *p*-values to determine significance when datasets become larger. For example, using a dataset with 100 values, of which 5 are really significant, a *p*-value of 0.05 would mean there are 5 false positives detected among 95 non-significant values, thus corresponding to a false discovery rate of 50% among the ten positives. The false discovery rate climbs when either the number of really significant values decreases or the number of non-significant values increases, the latter being a direct consequence of using large datasets such as those produced by RNA-Seq. In our study, when the FDR was fixed at 0.1, seven genes with significant difference were found. However, the number of significant differences decreases to 1 using a more stringent FDR of 0.05. It has been shown that the number of false positives recovered is considerably higher than the number expected [28]. This would agree with our observation that only two thirds of the DEG tested by qPCR were also found to show significant differences. Thus, in the light of the small number of significant changes found in our RNA-Seq experiment, we suggest that there are likely no real significant changes in transcript levels brought about by the changes in light intensity in our experiment. This would then agree with the lack of significant changes in transcript abundance over the course of the daily LD cycle using the dinoflagellate *L. polyedra* [18].

Our RNA-Seq experiment indicating there are no light induced transcripts has methodological differences with other reports in the literature suggesting the opposite. For example, an RNA-Seq study with *Symbiodinium microadriaticum* that showed 67 DEG when day and night were compared using DESeq with a false discovery rate (FDR) of 0.1 [16] used single samples rather than triplicate samples (Table 2). When we perform DESeq with an FDR = 0.1 using only one of three samples for each of the two time points, DESeq recovers 55 DEG instead of the seven DEG found when triplicate samples are used. Thus, in the *S. microadriaticum* study, insufficient replication may have exaggerated the number of light responsive transcripts. Another RNA-Seq study using *Symbiodinium* strain SSB01 looked at the number of DEG 24 h after a transfer from light to dark [17]. Here, 1334 DEG were found using cells grown phototrophically and 1739 DEG when cells were grown mixotrophically. These studies used duplicates (phototrophic growth) or triplicates (mixotrophic growth), but, instead of an FDR = 0.1, the cut-off values for significance were *p* < 0.05 and a fold change >1.5-fold. In our experiment, using triplicate samples with a similar cut-off value would result in 789 DEG instead of seven. Thus, the *Symbiodinium* SSB01 study had an exaggerated number of DEG because the cut-off criteria were not as stringent as using an FDR of 0.1. Both replicated samples and appropriate statistical analysis of significance are required for correct interpretation of RNA-Seq data.

It is important to emphasize that we do not propose dinoflagellates are incapable of transcriptional responses. However, in view of the experiments reported here, we believe it may be worthwhile re-examining the transcriptional response of dinoflagellates to stimuli other than light. A logical prediction from the permanently condensed chromatin that characterises dinoflagellate chromosomes is that transcriptional regulation is likely to be more difficult than in other cells. We thus suggest it may be important to verify transcriptional responses observed by a single method by using a complementary technique. Certainly, the finding of a true transcriptional response will be an important part in dissecting the molecular machinery that underpins this process in the dinoflagellates.

## Figures and Tables

**Figure 1 microorganisms-07-00261-f001:**
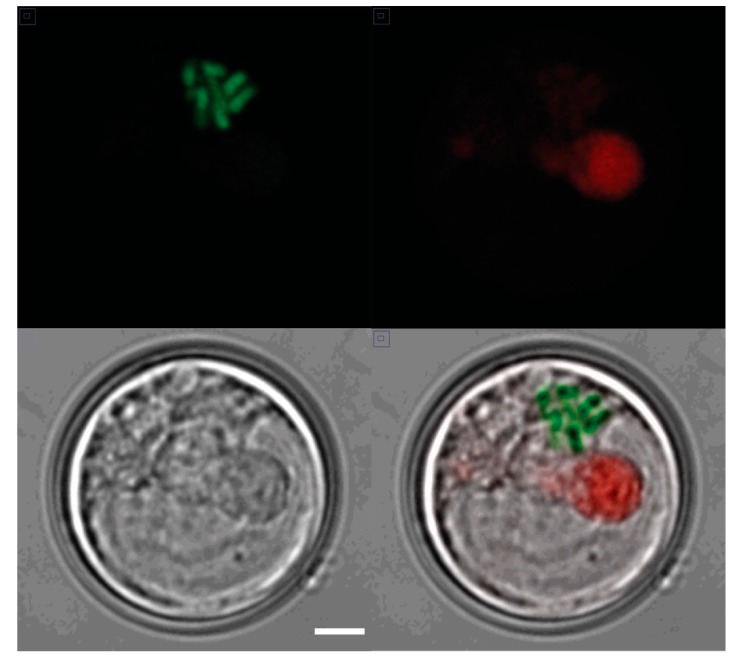
Condensed chromosomes in an interphase *S. kawagutii* cell. A confocal image of a single cell taken in (upper left) the green channel (PI staining of chromosomes), (upper right) the red channel (natural chlorophyll fluorescence), and (lower left) a DIC (Nomarski) image (scale bar 1 µm for all panels). A merged image is shown in the lower right.

**Figure 2 microorganisms-07-00261-f002:**
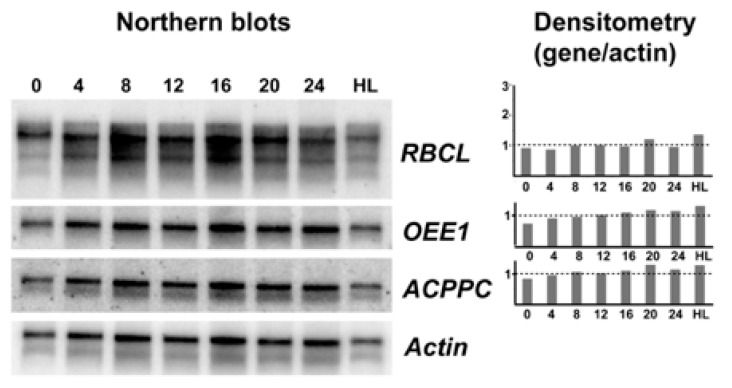
Northern blot analysis of three potentially light regulated transcripts. A representative sample of northern blots (*n* = 4) using either an *rbcL*, an *oee1*, an *acppc*, or an Actin cDNA as a probe. RNA was prepared from samples taken every four hours from cells grown under a normal 12:12 LD cycle as well as from cells grown under high light (note that LD 0 and LD 24 should be identical). At right, densitometric scans for the top three probes are shown relative to the Actin signal.

**Figure 3 microorganisms-07-00261-f003:**
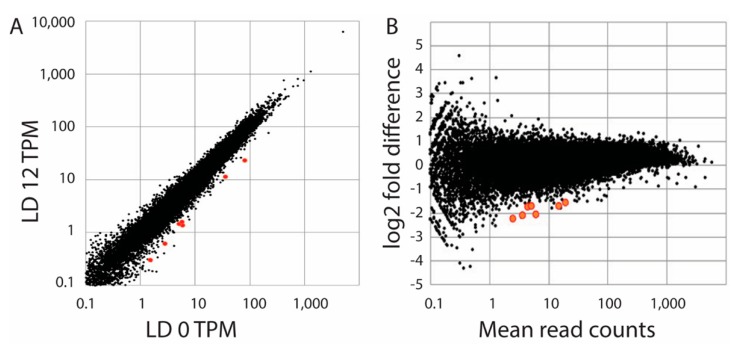
Comparison of transcript levels at LD 0 and LD 12. (**A**) A plot of read counts (as TPM, or transcripts per million) as the average of three samples at LD 12 are compared with the average of samples at LD 0. (**B**) An MA plot (fold-difference as a function of mean read count) is shown for triplicate samples at each of the two times as determined by DESeq2. The 7 sequences determined to be significantly different (*p*-adjust < 0.05; FDR = 0.1) are shown in red in both plots and are higher at LD 0 than at LD 12.

**Figure 4 microorganisms-07-00261-f004:**
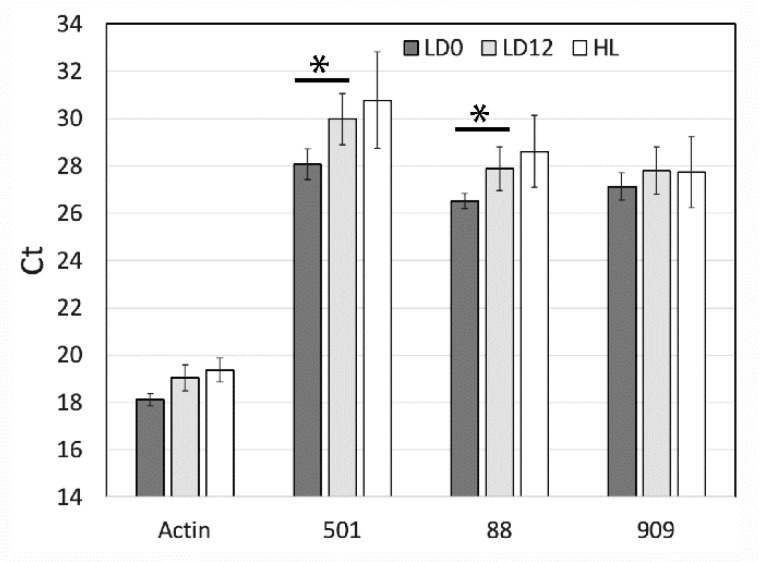
qPCR analysis of 3 selected light-regulated genes from RNA sequencing analysis. Ct values were obtained for three RNA-Seq predicted regulated genes (501, SymkaALLUN13501; 88, SymkaALLUN19088; 909, SymkaALLUN64909) as well as Actin as a control for the amount of cDNA. Triplicate samples from each of three biological replicates were averaged for LD 0 (samples were in the dark for 12 h), LD 12 (samples were in the light for 12 h), and for samples kept under constant high light for 24 h. Comparisons marked with * are significant at *p* < 0.01 using student’s *t*-test, respectively.

**Figure 5 microorganisms-07-00261-f005:**
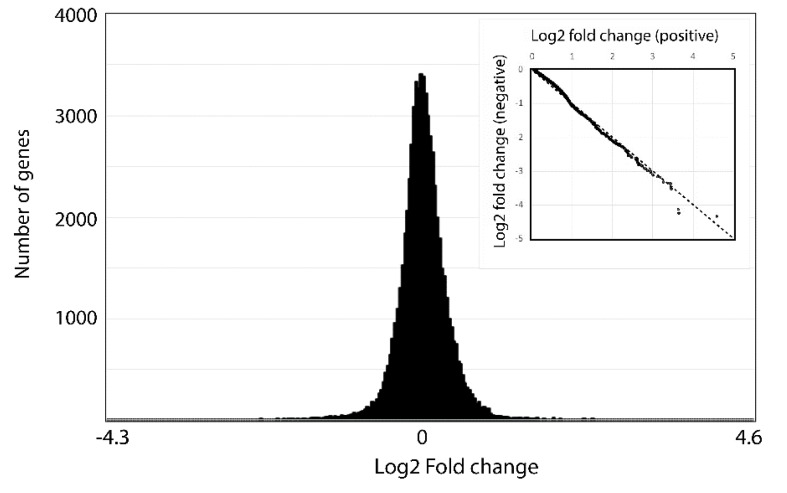
Fold changes are equally distributed. A histogram showing of the number of times a given log2 fold change is found in the data shows a normal distribution. The inset, showing a plot of the positive vs. negative log2 fold changes, is essentially an unbiased straight line indicating few genes are differentially expressed at one of the two times.

**Table 1 microorganisms-07-00261-t001:** Best BLAST hit for the seven potentially light regulated genes identified (False Discovery Rate FDR = 0.1).

Gene ID	Best BLAST Hit	E-Value	Fold Change
SymkaALLUN26766	aminomethyl transferase family protein [Halobellus limi]	1.6	0.33
SymkaALLUN13501	putative alanine aminotransferase, mitochondrial	3 × 10^−13^	0.23
SymkaALLUN70319	Hypothetical	9.7	0.24
SymkaALLUN19088	putative E3 ubiquitin-protein ligase HERC1	2 × 10^−21^	0.2
SymkaALLUN64909	LysM domain-containing protein	2.9	0.3
SymkaALLUN23766	No Sig Hits	-	0.3
SymkaALLUN19996	No Sig hits	-	0.29

**Table 2 microorganisms-07-00261-t002:** Differentially expressed genes (DEG) identified in different dinoflagellates after different treatments.

Species	Method	Comparisons	Replicates	FDR	*p*-Value	DEG	Reference
*S. kawagutii*	Illumina/DESeq2	LD0/LD12	3	0.05		1	This study
	Illumina/DESeq2	LD0/LD12	3	0.1		7	This study
	Illumina/DESeq2	LD0/LD12	3		<0.05	789	This study
	Illumina/DESeq2	LD0/LD12	1	0.1		55	This study
*S. microadriaticum*	Illumina/DESeq	LD0/LD12	1	0.1		67	[16]
	Illumina/DESeq	Normal/4 h 4 °C	1	0.1		119	[16]
	Illumina/DESeq	Normal/4 h 36 °C	1	0.1		2465	[16]
	Illumina/DESeq	Normal/12 h 34 °C	1	0.1		246	[16]
	Illumina/DESeq	Normal/4 h 20 g/L NaCl	1	0.1		138	[16]
	Illumina/DESeq	Normal/4 h 60 g/L NaCl	1	0.1		48	[16]
*Symbiodinium SSB01*	Illumina/DESeq	Light/24 h dark	3		<0.05	1334	[17]
*Symbiodinium*	Illumina/DESeq	29.2 °C/3 d 31.9 °C	2	0.05		0	[29]
	Illumina/DESeq	29.2 °C/3 d 31.9 °C	2		<0.05	541	[29]
*Symbiodinium* sp	Illumina/Student’s *t* test	Normal/4 d 31 °C	5	0.05		9471	[30]
	Illumina/Student’s *t* test	Normal/19 d 31 °C	5			12,701	[30]
	Illumina/Student’s *t* test	Normal/28 d 31 °C	5			13,269	[30]
*Lingulodinium polyedra*	Illumina/DESeq	Normal/1 d 4 °C	1	0.05		132	[31]
*Lingulodinium polyedra*	Illumina/DESeq	LD6/LD18	1	0.05		5	[18]
*Scrippsiella trochoidea*	Illumina/DESeq	Normal/N-limited	1	0.1		382	[32]
	Illumina/DESeq	Normal/P-limited	1	0.1		17	[32]
*Alexandrium tamarense*	MPSS/Fisher’s exact test	Normal/N-limited	1		<1 × 10^−10^	20	[33]
	MPSS/Fisher’s exact test	Normal/P-limited	1		<1 × 10^−10^	30	[33]
	MPSS/Fisher’s exact test	Normal/Xenic	1		<1 × 10^−10^	505	[33]
*Oxyrrhis marina*	454/Fisher’s exact test	30/50 practical saline units	1		<0.05	29	[34]
*Karenia brevis*	Microarray	Normal/N-limited	3		<0.0001	456	[35]
	Microarray	Normal/P-limited	3		<0.0001	425	[35]

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
