# Peer review of "Assessing Transcriptional Responses to Light by the Dinoflagellate Symbiodinium"

_microorganisms, 2019, doi:10.3390/microorganisms7080261_

Round 1
Reviewer 1 Report
Dinoflagellates are an important marine eukaryotic lineage with many ecological roles and unusual nuclear organization. Revealing gene regulation mechanisms at the transcriptional or translational level is important for understanding fundamental biological processes like symbiosis, parasitism and harmful algal bloom development. There has been conflicting information on whether dinoflagellates lack transcriptional control. This study combines 3 methods to show a lack of transcriptional response in a dinoflagellate during a light/dark cycle. Critically, it highlights the pitfalls of inadequate replication and data analysis stringency in previous dinoflagellate RNA-seq studies which have led to potentially erroneous conclusions about dinoflagellate gene transcription. In light of their findings, the authors suggest that previously published conclusions are re-evaluated. The manuscript is well-written, concise, and will be an important contribution to the field.
I agree with the overall approach and conclusions, if the data are indeed robust. Some additional information is needed to ensure the quality of the data. Since the study is using method comparisons to critically evaluate their findings, as well as those of other published papers, the technical approach needs to be sound and carefully explained. I suspect the authors have already performed the quality control analyses and just need to report them.
The authors should specify the quality controls of their RNA extracts in the Methods section. Did they assess RNA integrity, was it high and even among samples? Uneven RIN could mask small changes in expression. Also, was spectrophotometric assessment undertaken to confirm the purity of the sample? Since a phenol-based method was used for RNA extraction, the 260/230 ratio should be assessed to ensure no interference with cDNA synthesis. Was the RNA-seq done in triplicate for all samples? The authors state the reads recovered for six samples (i.e. no replicates,) on Line 105, but then report triplicates for LD0 vs LD12 on Line 150. The authors should clarify how many biological and technical replicates they have used for each method/analysis they have undertaken. Why was the actin gene chosen as a reference? Was is based on RNA-seq data, and did they check whether its expression was stable using a standard curve? The northern blot shows there may be variation in the expression. Since other genes are normalised to actin in northern blot and qPCR that would mask small changes in expression. For relative expression quantification, it is essential that the PCR amplification efficiency is identical for the reference gene and test genes, and generally higher than 0.9. Given that the number of genes studies is small, have the authors checked their qPCR efficiencies for the different primer pairs and what were they? The amount of cDNA used per reaction is quite high. I would recommend that ddPCR would be a better alternative for detecting small changes in transcript levels. If authors are able, they could check some of their samples, or at least mention it as a possible better method in the discussion. I am not sure of the value of the analysis presented in Figure 5. What does it tell us if the changes are not significant according to some statistic? Even “significant” changes shown in Table 1 are actually very small and unlikely to be real, as the qPCR has shown. I would suggest removing this section unless the authors can specify what it adds to their story. To strengthen the conclusion that other studies are retrieving false numbers of differentially expressed genes, e.g. as stated on line 239, I would recommend to process the raw reads from at least one of these published studies (e.g. SSB01, light:dark comparison) using the same workflow as the current study, and compare DEG retrieved using different thresholds.Specific comments in the text:
Line 87, include RNA-seq in the subtitle Inconsistent use of RNA-seq vs RNA seq. Please stick to one version. Inconsistent use of DeSeq2 and DeSeq (e.g. Line 111 vs Table 2, referring to this study). These work differently so please make sure the correct one is referred to. Line 93, please specify whether cDNA was produced using OligodT or hexamers; it is important when comparing RNA-seq to qPCR but unclear from this sentence. Line 110, please mention the read mapping tool. Also, an indication of whether mapped library size was similar among samples would be helpful.Author Response
Dinoflagellates are an important marine eukaryotic lineage with many ecological roles and unusual nuclear organization. Revealing gene regulation mechanisms at the transcriptional or translational level is important for understanding fundamental biological processes like symbiosis, parasitism and harmful algal bloom development. There has been conflicting information on whether dinoflagellates lack transcriptional control. This study combines 3 methods to show a lack of transcriptional response in a dinoflagellate during a light/dark cycle. Critically, it highlights the pitfalls of inadequate replication and data analysis stringency in previous dinoflagellate RNA-seq studies which have led to potentially erroneous conclusions about dinoflagellate gene transcription. In light of their findings, the authors suggest that previously published conclusions are re-evaluated. The manuscript is well-written, concise, and will be an important contribution to the field.
I agree with the overall approach and conclusions, if the data are indeed robust. Some additional information is needed to ensure the quality of the data. Since the study is using method comparisons to critically evaluate their findings, as well as those of other published papers, the technical approach needs to be sound and carefully explained. I suspect the authors have already performed the quality control analyses and just need to report them.
The authors should specify the quality controls of their RNA extracts in the Methods section. Did they assess RNA integrity, was it high and even among samples? Uneven RIN could mask small changes in expression. Also, was spectrophotometric assessment undertaken to confirm the purity of the sample? Since a phenol-based method was used for RNA extraction, the 260/230 ratio should be assessed to ensure no interference with cDNA synthesis.
For sequencing, the RNA was assessed by 28S/18S (Bioanalyser) and by 280/260 and 260/230 ratios (Nanodrop). The quality of all other RNA samples was assessed spectrophotometrically.
Was the RNA-seq done in triplicate for all samples? The authors state the reads recovered for six samples (i.e. no replicates,) on Line 105, but then report triplicates for LD0 vs LD12 on Line 150.
The six samples referred to are our two time points each performed in triplicate; this is now clarified in Methods. The Northern blots in figure 2 were performed four times (stated in the legend). The qPCR used three technical replicates from each of three biological replicates (stated in the legend).
The authors should clarify how many biological and technical replicates they have used for each method/analysis they have undertaken. Why was the actin gene chosen as a reference? Was is based on RNA-seq data, and did they check whether its expression was stable using a standard curve? The northern blot shows there may be variation in the expression. Since other genes are normalised to actin in northern blot and qPCR that would mask small changes in expression.
We have previously used actin as a reference for another species (Lingulodinium polyedra) as levels do not change by either Northern blots or RNA-Seq. Actin also shows no variation in levels by RNA-Seq with Symbiodinium.
For relative expression quantification, it is essential that the PCR amplification efficiency is identical for the reference gene and test genes, and generally higher than 0.9. Given that the number of genes studies is small, have the authors checked their qPCR efficiencies for the different primer pairs and what were they? The amount of cDNA used per reaction is quite high. I would recommend that ddPCR would be a better alternative for detecting small changes in transcript levels. If authors are able, they could check some of their samples, or at least mention it as a possible better method in the discussion.
We did not measure the amplification efficiency; since our experiments showed little change in Ct values at the different times for any given primer pair, we reasoned no correction for differences in RNA levels between samples would be required. To make this clear, we have altered the figure to present Ct values directly instead of DCt. We did ensure that only a single band of the correct size was amplified by each primer pair. We are not currently able to perform digital PCR, but note that the magnitude of the changes we were looking for were not small - based on RNA-Seq we were expecting between three to five fold.
I am not sure of the value of the analysis presented in Figure 5.
In our view this was an original way to globally present a lack of gene transcript levels; a perfect bell curve (equal numbers of induced and repressed genes) would be represented by a diagonal line. If this analysis is unclear we can move this figure to supplementary data.
What does it tell us if the changes are not significant according to some statistic? Even “significant” changes shown in Table 1 are actually very small and unlikely to be real, as the qPCR has shown. I would suggest removing this section unless the authors can specify what it adds to their story. To strengthen the conclusion that other studies are retrieving false numbers of differentially expressed genes, e.g. as stated on line 239, I would recommend to process the raw reads from at least one of these published studies (e.g. SSB01, light:dark comparison) using the same workflow as the current study, and compare DEG retrieved using different thresholds.
Statistical analysis for significance is always reported, and while it may not always indicate real results, it is an essential element in evaluating and comparing different results. We agree that the changes shown in Table 1 are actually small but also think they should be reported.
While we did not reprocess other datasets, we did analyse our own data with different thresholds for significance and differing degrees of replication. These results were included in table 2.
Specific comments in the text:
Line 87, include RNA-seq in the subtitle Inconsistent use of RNA-seq vs RNA seq. Please stick to one version. Inconsistent use of DeSeq2 and DeSeq (e.g. Line 111 vs Table 2, referring to this study). These work differently so please make sure the correct one is referred to. Line 93, please specify whether cDNA was produced using OligodT or hexamers; it is important when comparing RNA-seq to qPCR but unclear from this sentence. Line 110, please mention the read mapping tool. Also, an indication of whether mapped library size was similar among samples would be helpful.
Corrections have been made to the text to accommodate these issues. The reads were mapped and counted using Salmon. The number of mapped and counted reads was very similar between the samples.
Reviewer 2 Report
This is a detailed study done by a well experienced lab, on an important, but unappreciated group of organisms. The researchers have applied modern biological methods to investigate an very interesting, but extremely interesting alternative to the common format of Eukaryotic gene regulation at the transcriptional level. The authors proved their point that our understanding of the regulation of transcription by light is still poorly understood.
The researchers interests are important, and the data is sound. As a "typo alert", I wonder if Reference 7 (line 274), was really published RNA spelled in lower case. On a more scientific note, I would like to address possible "false positives" in the area of the regulation of transcription in dinoflagellates. I am not up on the literature on this topic, so it may have already been addressed. A good control for these kids of studies would be to test the "positive results" on a nonphotosynthetic dino like C. chonii. Any positives that show up in a nonphotosynthetic should be questioned, even if there are light induced transcriptions that do not involve photosynthsis.
Author Response
This is a detailed study done by a well experienced lab, on an important, but unappreciated group of organisms. The researchers have applied modern biological methods to investigate an very interesting, but extremely interesting alternative to the common format of Eukaryotic gene regulation at the transcriptional level. The authors proved their point that our understanding of the regulation of transcription by light is still poorly understood.
The researchers interests are important, and the data is sound. As a "typo alert", I wonder if Reference 7 (line 274), was really published RNA spelled in lower case. On a more scientific note, I would like to address possible "false positives" in the area of the regulation of transcription in dinoflagellates. I am not up on the literature on this topic, so it may have already been addressed. A good control for these kids of studies would be to test the "positive results" on a nonphotosynthetic dino like C. chonii. Any positives that show up in a nonphotosynthetic should be questioned, even if there are light induced transcriptions that do not involve photosynthsis.
The idea is very important, and I firmly believe some reliable method for weeding out false positives due to any stimuli should be developed. As far as I can tell, this issue has NOT been adequately dealt with in the dinoflagellate literature. However, the two potential issues I see when comparing light responses of genes in photosynthetic and non-photosynthetic species would be (1) gene expression may be influenced by photoreceptors and (2) that the gene in question may not be present in both species.
Round 2
Reviewer 1 Report
I am glad that the authors already had all the relevant controls for their data. I am satisfied with the clarifications the authors have provided and with their edits in the manuscript, and recommending acceptance of the manuscript.